

# Impacts of forest age on soil characteristics and fertility quality of *Populus simonii* shelter forest at the southern edge of the Horqin Sandy Land, China

Xinyu Guo[1], Guang Yang[1], Ji Wu[1], Shi Qiao[1] and Li Tao[2]

[1] College of Desert Control Science and Engineering, Inner Mongolia Agricultural University, Hohhot, Inner Mongolia, China
[2] Forest General Site of Ordos City, Ordos, Inner Mongolia, China

Corresponding author
Guang Yang, yg331@126.com

## ABSTRACT

The sand fixing shelter forests in the Horqin Sandy Land are a key area in the "3-North" Shelter Forest Program in China, which has a history of over 50 years of artificial afforestation. *Populus simonii* Carr is one of the most dominant silvicultural species in the region. The aim of this study is to understand the soil characteristics and soil fertility of *Populus simonii* shelter forests at different growth stages and to establish a scientific basis for soil nutrient regulation and sustainable management of *Populus simonii* shelter forests at the southern edge of the Horqin Sandy Land. Sample plots were selected for young (≤15 a), middle-aged (16–25 a), near-mature (26–30 a), mature (31–40 a), and over-mature (≥41 a) forests. Each forest studied was in a state of natural restoration with uniform stand conditions and no artificial fertilizer was applied. These sites were selected to study changes in the soil characteristics in soil depths of 0–20, 20–40, and 40–60 cm. In order to avoid the problem of multicollinearity between soil variables and to reduce redundancy, principal component analysis (PCA), Pearson's correlation analysis, and Norm value calculation were used to select the least correlated indicators with the highest factor loadings. This was used to establish the minimum data set. The soil fertility quality of these shelterbelts in different forest ages was quantified using the soil quality index (SQI). In the growth stage from young to nearly mature forests, the soil bulk weight and pH decreased with increasing forest age. Soil capillary porosity, noncapillary porosity, total porosity, water content, field water holding capacity, and organic carbon content increased with increasing forest age and soil nutrient content gradually improved. At the stage of near-mature to over-mature forests, the effect of forest age on soil bulk density was not significant and all other soil characteristics decreased to varying degrees as the forest age increased. The soil also developed from alkaline to neutral. The SQI of the total data set and the SQI of the minimum data set consistently showed that near-mature forests (NMF) > middle-aged forests (MAF) > mature forests (MF) > over-mature forests (OMF) > young forests (YF). The results of the two evaluation systems showed a significant positive correlation ($P < 0.05$, $R^2 = 0.8263$) indicating that it is feasible to use the minimum data set to evaluate the soil fertility of shelter forests of different forest ages. The age of the forest has an obvious effect on the soil characteristics and overall soil fertility of shelter forests. The *Populus simonii* shelter forests on the southern edge of the Horqin Sandy Land

have great soil development at the early stage of afforestation and the soil nutrient content gradually increases. The soil fertility reaches a peak when the forest is nearly mature and the soil fertility declines after the age of the forest reaches 30 years.

# INTRODUCTION

From the 1970s to the present, the Chinese government has carried out large-scale artificial forestry and ecological projects in Northwest China, North China, and Northeast China in order to combat desertification and improve the ecological environment. This project is known around the world as the "3-North" Shelter Forest Program. As a key project area of the "3-North" Shelter Forest Program, the Horqin Sandy Land has since created large-scale sand fixing shelter forests and *Populus* has become the main afforestation tree species in the area. Numerous studies have demonstrated the high water utilization of *Populus* shelterbelts (*Fu, Sun & Luo, 2016*). These shelterbelts prolong the retention time of soil organic carbon by reducing soil Q10 (*Zhang et al., 2022*), alter the physico–chemical properties and microbial population of the soil, and increase the diversity of the vegetation (*Jiang et al., 2013*). The study area of this study is located in Naiman County, Tongliao City, Inner Mongolia, at the southern edge of the Horqin Sandy Land. The area of *Populus* artificial forests within Tongliao City exceeds 1/3 of the total area of artificial forests in Inner Mongolia and represents up to 1/10 of the area of *Populus* artificial forests in China. *Populus simonii* is the most widely planted *Populus* species in the shelter forests in the study area (*Yan, 2016*). As the number of years of afforestation increases, the problems of potential soil fertility retrogression, low nutrient recycling rate, and degradation of middle-aged and old-growth forests are becoming more and more obvious in *Populus simonii* shelter forests (*Cao, 2008*; *Ni, 2017*; *Shi et al., 2022*). The influence of physico-chemical properties of soil, such as water content, water holding capacity, and nutrients, on forest stand decline has received much attention from scholars (*Abaker et al., 2018*; *Liang et al., 2022*; *Wardle, Walker & Bardgett, 2004*).

Soil, as the basis for plant growth and survival, plays an indispensable role in the growth and development of trees and the structure and function of plant communities (*Jia et al., 2020*; *Normand et al., 2017*). The creation of planted forests significantly alters the soil properties of afforestation sites (*Wu et al., 2018*) whose soil physico-chemical properties in turn influence the species composition of individual plants and understory vegetation in forest communities (*Maiti & Ghosh, 2020*; *Wang & Zheng, 2020*). Soil evaluation provides a quantitative analysis of the total soil properties (*Doran & Zeiss, 2000*). With the development and integration of multiple disciplines when conducting an evaluation of the soil, the specific methodology of the evaluation has advanced from traditional qualitative analyses (*Karlen, Ditzler & Andrews, 2003*) to diversified quantitative analyses (*Askari & Holden, 2014*). This is often expressed through the biological and physico-chemical properties of the soil (*Bhardwaj et al., 2011*). Common soil quality evaluation methods

include fuzzy correlation (*Zhang et al., 2017*), geographic information systems (*AbdelRahman, Shalaby & Mohamed, 2019*), and soil quality indices (*Ma et al., 2022*). Among these methods, the soil quality index method is widely used because of its ease of calculation and high degree of quantitative adjustability (*Hammac et al., 2016*).

In recent years, a number of studies on the physico-chemical properties of soils in artificial forests of different ages have proved that the age of the forest has a direct effect on soil characteristics (*Ali et al., 2019*; *Fonseca, Alice & Rey-Benayas, 2012*; *Lucas-Borja et al., 2019*; *Niu et al., 2015*; *Yesilonis et al., 2016*; *Zeng et al., 2017*). The results of the comprehensive evaluation of soils of different forest ages have also been frequently reported (*Chen et al., 2021b*; *Cao et al., 2023*; *Fuss et al., 2019*; *Shi et al., 2016*). Many scholars have studied soils of artificial forests of different forest ages mainly in conventional conditions but there are not many relevant studies in arid and semi-arid areas. Sand shelter forests fulfil a unique ecological function and are rarely harvested, even at maturity. Sandy soils are highly permeable, making them not only poor at holding water but also unable to store and exchange nutrients well (*Huang & Hartemink, 2020*). Declining soil fertility not only seriously hinders the growth and development of above-ground vegetation but also affects soil and water conservation and wind and sand stabilization of shelter forests (*Zhang & Huisingh, 2018*). Therefore, determining how the soil is developing in shelter forests at different growth stages is important for maintaining soil nutrient levels and for sustainable management of shelter forests in the study area. The objectives of this study were to test the following hypotheses: Assuming that the understory soil properties of *Populus simonii* shelter forests in sandy areas under natural restoration conditions are affected by the age of the forest we hypothesized that the physico-chemical properties and soil fertility of the understory soils at different growth stages do not always increase but rather they fluctuate. We hypothesize that such changes are different at varying soil depths.

## MATERIALS AND METHODS

### The study area

The study area of this study is located in Naiman County, Tongliao City, in the Inner Mongolia Autonomous Region of China and physically located at the southern edge of the Horqin Sandy Land in China (120°19′40″–121°35′40″E, 42°14′40″–43°32′20″N). The area is 8,137.6 km. In the 1970s, 62% of the total study area was sandy soil. The government began large-scale afforestation measures in the Horqin Sandy Land. The area has now reached 30.45% forest cover. The climate of the study area is northern temperate and arid with a continental monsoon. Elevation ranges from 250 to 570 m. The average annual temperature is 6.5–7.7 C. The average annual wind speed is 3.6–4.1m/s. The average annual precipitation is 274–344 mm. The average annual evaporation is 1,817.4–2,032.6 mm and is about six times the average precipitation. According to the *FAO (1988)*, soil texture in the sampling area is mainly Arenosols. The main herbaceous vegetation under the *Populus simonii* shelter forests is *Lespedeza davurica*, *Setaria viridis*, *Salsola collina*, and *Bassia scoparia*. The study area is shown in Fig. 1.

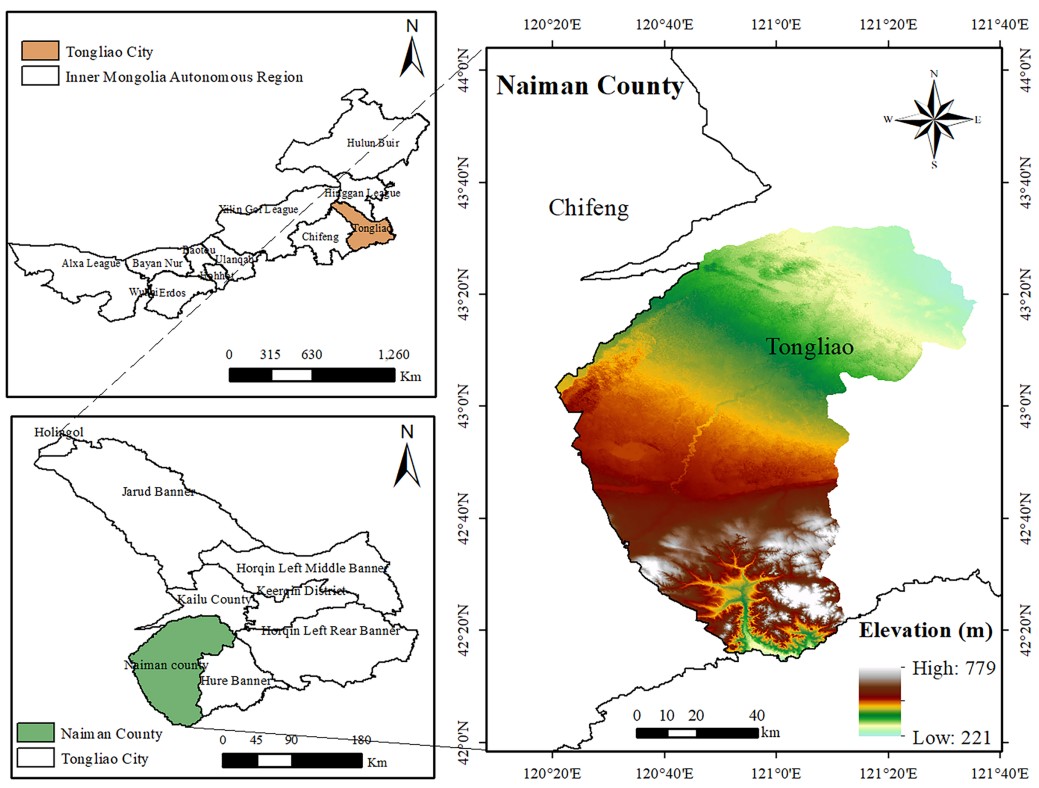

**Figure 1 Location map of the study area.** Note: Plotting was done using Arcgis 10.6. The map data set is provided by Geospatial Data Cloud site, Computer Network Information Center, Chinese Academy of Sciences. (http://www.gscloud.cn).

## Study methods

### Sample plot setting

In this study, we refer to the current Chinese National Forestry Industry Standard "Age Classification and Age Group Division of Major Tree Species" (*PRC State Forestry Administration, 2017*) for the age classification of planted poplar protection forests in northern China. Three sample plots each of young, middle-aged, near-mature, mature and over-mature forests were selected from afforestation areas in the study area, This resulted in a total of 15 sample plots. The types of shelterbelts selected in this study were all sand stabilization forests. The selected plots had the same site conditions, similar initial afforestation densities, and all plots were in a state of natural restoration without artificially applied fertilizer. Each sample area was 0.09 hm² (30 m * 30 m). In each sample plot 15 standard trees were selected to calculate the average tree height. Tree heights were measured using an altimeter. The basic information of the *Populus simonii* shelter forests sample plots is shown in Table 1.

### Soil sample collection and determination

Soil samples were collected from August 23rd–27th, 2022. Three soil sampling points were randomly selected on the diagonal of the sample plot and a soil pit with a depth of 60 cm was dug out at the sampling point. Samples were taken from the bottom to the top of the

**Table 1 Basic information of the sample plots.**

| Age class | Division of age class (a) | Symbol | Forest age (a) | Initial afforestation density (trees·hm$^{-2}$) | Mean tree height (m) |
|---|---|---|---|---|---|
| Young forest | ≤15 | YF | 12 | 1,089 | 7.93 ± 0.43 |
| | | | 14 | 1,101 | |
| | | | 15 | 1,101 | |
| Middle-aged forest | 16–25 | MAF | 20 | 1,122 | 9.76 ± 0.84 |
| | | | 21 | 1,122 | |
| | | | 22 | 1,122 | |
| Near-mature forest | 26–30 | NMF | 27 | 1,122 | 11.64 ± 0.67 |
| | | | 29 | 1,122 | |
| | | | 30 | 1,141 | |
| Mature forest | 31–40 | MF | 35 | 1,144 | 14.08 ± 0.63 |
| | | | 35 | 1,144 | |
| | | | 36 | 1,144 | |
| Over-mature forest | ≥41 | OMF | 41 | 1,205 | 16.30 ± 0.71 |
| | | | 41 | 1,205 | |
| | | | 42 | 1,205 | |

**Note:**
Data are mean ± standard deviation.

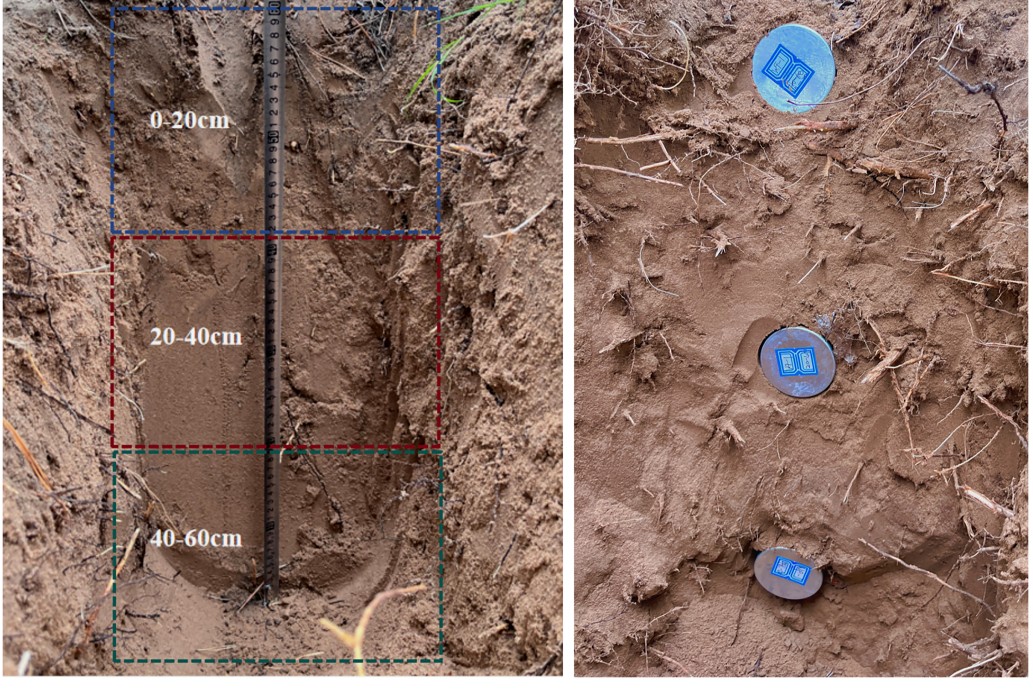

**Figure 2 Schematic diagram of soil sampling.**

soil profile in three layers (0–20, 20–40, 40–60 cm). The sampling method is shown in Fig. 2. Soil water content, bulk density, field water holding capacity, and porosity were collected from undisturbed soil with a 100 cm$^3$ ring knife. Samples were dried at 105 °C for

12 h and analyzed using the gravimetric method. Samples to determine the soil chemical properties were brought back to the laboratory in sealed bags and were air-dried in the absence of sunlight, All soil was divided into two portions; one portion was passed through a 1 mm soil sieve in order to determine the available nutrients content, the other portion was passed through a 0.25 mm soil sieve in order to determine the total nutrients content. Soil pH was measured using the potentiometric method. Soil organic carbon was measured using the volumetric heating method with potassium dichromate. Available nitrogen was measured using the alkali-diffusion method. Available phosphorus was measured by 0.5 mol/L $NaHCO_3$-$H_2SO_4$ leaching using a molybdenum antimony anti-colourimetric assay. Available potassium was measured using the 1 mol/L $CH_3COONH_4$ leaching-flame photometric method. Total nitrogen was measured using the Kjeldahl method with a concentrated sulphuric acid-perchloric acid mixture. Total phosphorus was measured by concentrated sulphuric acid-perchloric acid mixed cooking-molybdenum antimony anti-colourimetric method. Total potassium was measured using the acid dissolution-flame photometric method. This experimental study covered five age classes. Each age class had three sample plots. Three layers were sampled in each sample plot. Three soil samples were collected in each soil layer. A total of 135 soil samples were collected.

### Soil fertility assessment methods

In order to make the evaluation indicators of the soil under different shelterbelts comparable, the data of the soil property indicators was normalized according to the principle of fuzzy mathematics. An affiliation function for soil properties and functions was established. The general affiliation function was divided into parabolic, trapezoidal, and "S" shapes according to the characteristics of the soil and the different effects of the corresponding functional realization. Referring to the division of membership degrees of soil evaluation indicators in previous studies (*Jahany & Rezapour, 2020*; *Nabiollahi et al., 2017*; *Wang et al., 2020*), it was determined that soil porosity, soil water content, organic carbon and total nitrogen are membership functions of "S" shaped curves and soil bulk density is a membership function of parabolic curves.

"S" type membership function is:

$$F(x_i) = (x_i - x_{i\,min})/(x_{i\,max} - x_{i\,min}). \tag{1}$$

Parabola type membership function is:

$$F(x_i) = (x_{imax} - x_i)/(x_{i\,max} - x_{i\,min}) \tag{2}$$

where: $F(X_i)$ is the membership value of each soil evaluation indicator; $X_i$ is the value of each soil indicator; $X_{imax}$ is the maximum value in the soil indicator of item $i$; and $X_{imin}$ is the minimum value in the soil indicator of item $i$.

Soil fertility is influenced by soil nutrients including nitrogen, phosphorus, and potassium, which are macronutrients found in the soil. Soil nutrients are affected by organic carbon and pH levels and nutrient uptake and transformation is effected by the

soils physical properties (*Gao & Huang, 2020*; *Lukina, Orlova & Isaeva, 2011*; *Marage & Gégout, 2009*). Fourteen soil indicators that can affect the soil fertility characteristics were selected to establish the total data set (TDS) for evaluating the soil fertility of shelter forests. These included soil water content, bulk density, noncapillary porosity, capillary porosity, total porosity, field water holding capacity, soil pH, organic carbon, available phosphorus, available potassium, available nitrogen, total nitrogen, total phosphorus, and total potassium.

The minimum data set (MDS) was proposed by *Larson & Pierce (1991)* and it has been widely used in soil quality assessment to avoid the problem of multicollinearity between soil variables and to reduce redundancy. The MDS was constructed by combining Pearson correlation analysis and Norm value through principal component analysis. The principal component vectors with an eigenvalue of the principal component matrix greater than 1 and a cumulative contribution rate greater than 80% were extracted. The main influencing factors in the principal components were determined according to the factor loadings. The indicators with a high Norm value in each set of principal components were selected. The correlation between the indicators was compared. The indicators used in the MDS were those with relatively high Norm values and relatively small correlation between the indicators.

The Norm value is:

$$N_{ik} = \sqrt{\sum_1^k \left( u_{ik}^2 \lambda_k \right)} \tag{3}$$

where: $N_{ik}$ is the integrated loading value (Norm value) on soil indicator i on the first principal component, $U_{ik}$ is the loading value of soil variable i on the principal component $k$, $\lambda_k$ is the eigenvalue corresponding to the principal component $k$, and $k$ is the number of principal components with an eigenvalue $\geq 1$.

The variance contribution ratio of each indicator of TDS and MDS was calculated separately to determine the weight of the indicators. The indicators that reflect soil fertility were normalized by the scale and the soil quality indicator method was used to calculate the composite score of the soil fertility.

SQI is:

$$SQI = \sum_{j=1}^n W_j \times S_j \tag{4}$$

where: $n$ is the total number of soil indicators extracted, $W_j$ is the weight of indicator $j$, and $S_j$ is the membership value of the evaluation indicator $j$.

### Data processing and analysis

PowerPoint was used to draw the schematic diagram of soil sampling in this article and Origin Pro 2021 and Excel were used for the other drawings. One-way ANOVA, LSD multiple comparisons, Pearson correlation analysis, and principal component analysis

(PCA) were performed using SPSS 26.0 software. This was done to test the significance of the differences between the soil indicator data at different soil depths for different forest ages and to determine the MDS indicators and the weights of the indicators in the soil fertility evaluation.

## RESULTS

### Changes in soil bulk density and porosity

Figure 3 shows the changes in soil bulk density and porosity of *Populus simonii* shelter forests of different forest ages. The soil bulk density in general showed the pattern of change as NMF < MF < OMF < MAF < YF. Soil bulk density at different soil depths of the same age class increased at each soil depth so that 0–20 < 20–40 < 40–60 cm. There was a significant difference in soil bulk density among YF, MAF and NMF ($p < 0.05$) and there was no significant difference between MF, OMF and MAF, NMF ($p < 0.05$). This indicated that when the shelter forest grows beyond MF the forest age no longer has a significant effect on the soil bulk density. The changes in the soil non-capillary porosity, capillary porosity, and total porosity was opposite that of soil bulk density. Specifically, we determined that: NMF > MF > OMF > MAF > YF. The porosity of different soil depths of the same age class showed 0–20 > 20–40 > 40–60 cm. Soil total porosity was significantly different ($p < 0.05$) among the sample plots of all age classes, indicating that the growth and development of artificial vegetation as the forest aged had a significant effect on soil porosity.

### Changes in soil water content and field water holding capacity

Changes in soil water content and field water holding capacity of *Populus simonii* shelter forests in different age classes are shown in Table 2. Soil water content and field water holding capacity in general both showed the pattern of change as NMF > MF > MAF > OMF > YF. Soil water content at different soil depths of the same age class varied as follows: 0–20 < 20–40 < 40–60 cm. Soil water content at the same age class was significantly different at all soil depths ($p < 0.05$). The variation of field water holding capacity in soil depth was 0–20 > 20–40 > 40–60 cm. The field water holding capacity of the 0–20 cm soil layer was significantly different from the water holding capacity at a soil layer depth of 20–40 and 40–60 cm ($p < 0.05$).

### Changes in soil pH and organic carbon content

Table 3 shows the changes in soil pH and organic carbon in shelter forests of different forest ages. Soil pH gradually decreased as forest age increased and the soil pH of MF and OMF was significantly different from that of the YF, MAF, and NMF sample plots ($P < 0.05$). This indicates that the soil shifted from alkaline to neutral as the forest aged. The soil organic carbon content of different age classes varied as NMF > MF > OMF > MAF > YF and it decreased with increasing soil depth. In the YF and MAF classes, organic carbon was significantly different ($P < 0.05$) between the 0–20 cm soil depth and the 20–40 cm, whereas there was no significant difference ($P > 0.05$) between the organic carbon content of the 20–40 and 40–60 cm soil depths. There was a significant difference

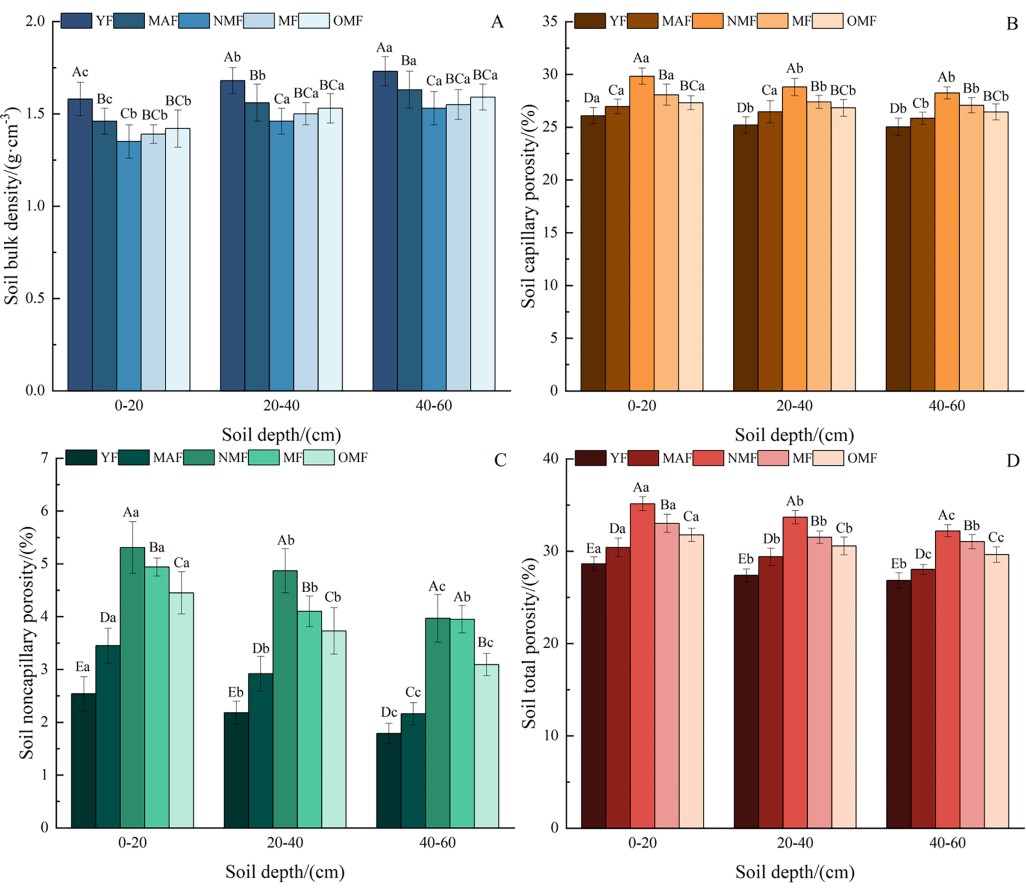

**Figure 3 Changes in soil bulk density and porosity.** Note: Different uppercase letters in the same soil depth indicate significant differences between different forest age classes while different lowercase letters in the same age classes indicate significant differences between different soil depths, ($p < 0.05$). YF, Young forest; MAF, Middle-aged forest; NMF, Near-mature forest; MF, Mature forest; OMF, Over-mature forest.

**Table 2 Changes in soil water content and field water holding capacity in shelter forests of different forest ages.**

| Soil characteristics | Soil depth/cm | YF | MAF | NMF | MF | OMF |
|---|---|---|---|---|---|---|
| Soil water content/% | 0–20 | 4.69 ± 0.23Ea | 5.33 ± 0.29Ca | 6.65 ± 0.24Aa | 6.10 ± 0.26Ba | 5.04 ± 0.21Da |
| | 20–40 | 6.19 ± 0.32Cb | 7.28 ± 0.20Bb | 7.80 ± 0.17Ab | 7.65 ± 0.17Ab | 7.07 ± 0.17Bb |
| | 40–60 | 7.50 ± 0.44Dc | 8.60 ± 0.10Bc | 9.27 ± 0.17Ac | 8.83 ± 0.24Bc | 7.98 ± 0.21Cc |
| Field water holding capacity/% | 0–20 | 24.86 ± 0.87Da | 25.86 ± 0.78Ca | 27.81 ± 0.49Aa | 26.53 ± 0.53Ba | 25.55 ± 1.26CDa |
| | 20–40 | 24.21 ± 1.22Db | 25.11 ± 0.93Cab | 26.89 ± 0.42Ab | 26.04 ± 0.31Bab | 24.98 ± 0.72Cb |
| | 40–60 | 23.86 ± 0.50Cc | 24.98 ± 0.57Bb | 26.33 ± 0.45Ab | 25.59 ± 0.98ABb | 24.82 ± 0.78Bb |

**Note:**
Different uppercase letters in the same soil depth indicate significant differences between different forest age classes while different lowercase letters in the same age classes indicate significant differences between different soil depths, ($p < 0.05$). YF, Young forest; MAF, Middle-aged forest; NMF, Near-mature forest; MF, Mature forest; OMF, Over-mature forest.

($P < 0.05$) between the organic carbons in each soil depth of the NMF, MF and OMF. This indicates that the changes in soil organic carbon content in the early stage of forest growth and development were mainly concentrated in the surface soil layer and the differences in organic carbon content among soil depths were more obvious after NMF.

**Table 3 Changes in soil pH and organic carbon in shelter forests of different forest ages.**

| Soil characteristics | Soil depth/cm | YF | MAF | NMF | MF | OMF |
|---|---|---|---|---|---|---|
| Soil pH | 0–20 | 7.38 ± 0.15Aa | 7.35 ± 0.07Aa | 7.31 ± 0.06Aa | 7.27 ± 0.08Ba | 7.25 ± 0.07Ba |
| | 20–40 | 7.27 ± 0.05Ab | 7.25 ± 0.05Ab | 7.22 ± 0.06Ab | 7.17 ± 0.07Bb | 7.15 ± 0.05Bb |
| | 40–60 | 7.20 ± 0.05Ab | 7.19 ± 0.05Ab | 7.13 ± 0.06Bc | 7.12 ± 0.07Bb | 7.09 ± 0.04Bb |
| Soil organic carbon/(g/kg) | 0–20 | 5.10 ± 0.62Ea | 5.63 ± 0.56Da | 8.79 ± 0.26Aa | 7.37 ± 0.4Ba | 6.02 ± 0.32Ca |
| | 20–40 | 4.14 ± 0.37Eb | 4.90 ± 0.42Db | 7.67 ± 0.26Ab | 6.61 ± 0.3Bb | 5.59 ± 0.29Cb |
| | 40–60 | 3.87 ± 0.33Db | 4.53 ± 0.34Cb | 6.99 ± 0.37Ac | 6.12 ± 0.18Bc | 4.91 ± 0.24Cc |

**Note:**
Different uppercase letters in the same soil depth indicate significant differences between different forest age classes while different lowercase letters in the same age classes indicate significant differences between different soil depths, ($p < 0.05$). YF, Young forest; MAF, Middle-aged forest; NMF, Near-mature forest; MF, Mature forest; OMF, Over-mature forest.

## Changes in soil nutrient content

Soil nutrient content indicators of different age classes of *Populus simonii* shelter forests all peaked during NMF (Fig. 4). The soil total nitrogen, total phosphorus, and available phosphorus content varied as a whole with NMF > MF > OMF > MAF > YF. Soil total potassium, available nitrogen, and available potassium (0–20 cm soil depth) were NMF > MAF > MF > OMF > YF. Soil nutrient content all decreased with increasing soil depth. ANOVA showed that there was a significant difference in the distribution of soil nutrients among the three soil depths during the NMF ($P < 0.05$) whereas the soil nutrient content of the MF and the OMF gradually decreased and was closer to the nutrient level of the MAF. This may be related to the competition for water and nutrients caused by the increase in vegetation cover in the later stages of forest growth and development.

## Evaluation of soil fertility quality

### Constructing the minimum data set

A total of two principal components with eigenvalues ≥1 were extracted through PCA (Table 4), with eigenvalues of 9.983 and 2.079. Their cumulative variance contribution rate was 86.159%, indicating that the two extracted principal components are representative of the comprehensive properties of each indicator. The results of the Pearson correlation analysis of the indicators are shown in Fig. 5. In accordance with the requirement that the absolute value of the principal component loadings of the soil indicators must be greater than 0.5, the indicators used in the MDS were those with relatively high Norm values and relatively small correlation between the indicators. Soil total phosphorus with the highest Norm value was included in the MDS at PC-1. Available phosphorus (correlation coefficient <0.6 with total phosphorus), which had a relatively large Norm value and a relatively small correlation coefficient with total phosphorus, was included in the MDS. The correlation coefficient between pH and soil water content in PC-2 was <0.55 and both were included in the MDS. In summary, the final indicators selected for inclusion in the MDS were soil total phosphorus, available phosphorus, water content, and pH.

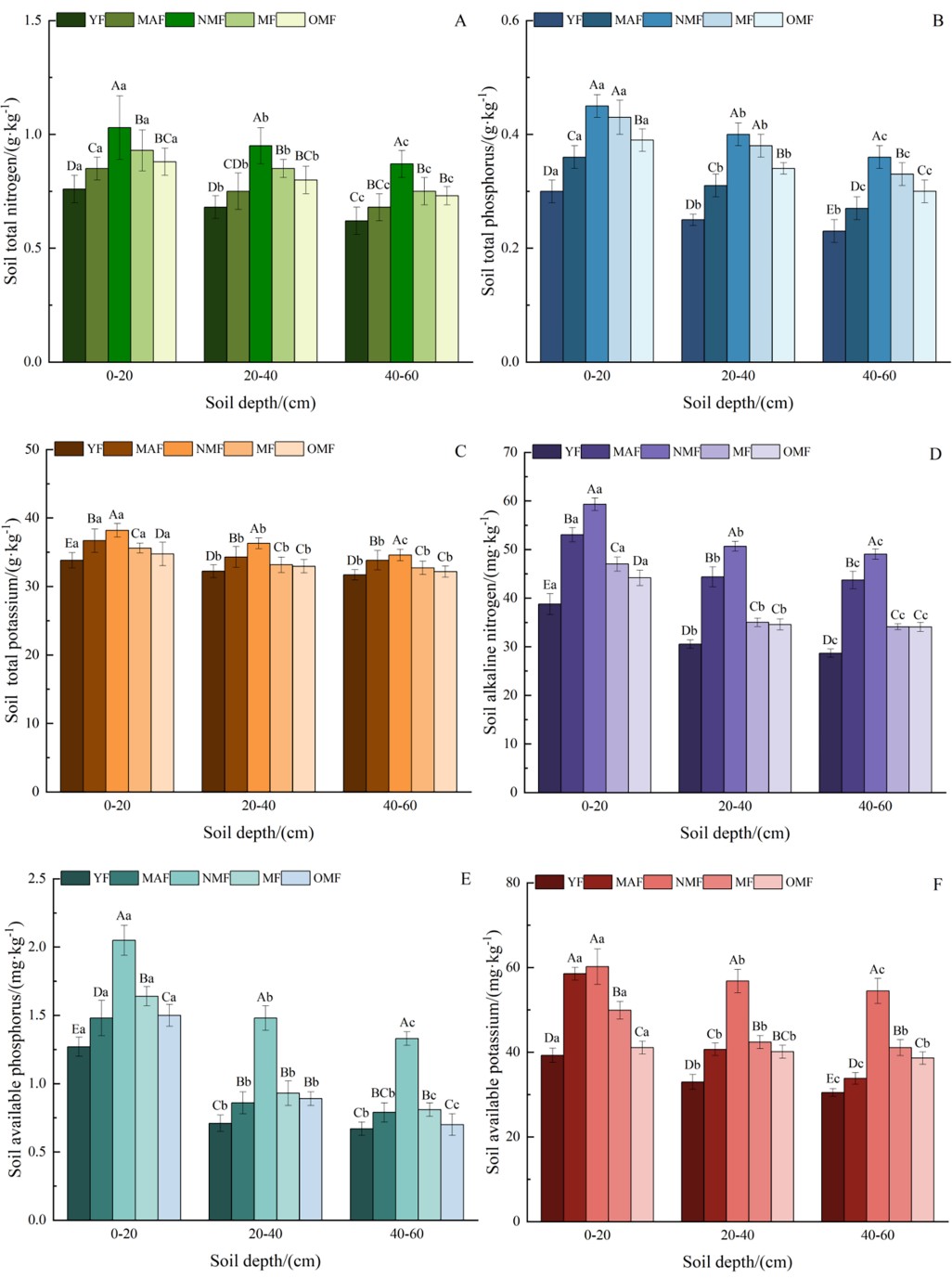

**Figure 4  Changes in soil nutrients.** Different uppercase letters in the same soil depth indicate significant differences between different forest age classes while different lowercase letters in the same age classes indicate significant differences between different soil depths, ($p < 0.05$). YF, Young forest; MAF, Middle-aged forest; NMF, Near-mature forest; MF, Mature forest; OMF, Over-mature forest.

**Table 4 Results of principal component analysis of soil quality indicators.**

| Soil indicator | Symbol | Principal component | | Norm value | Minimum date set |
|---|---|---|---|---|---|
| | | PC-1 | PC-2 | | |
| Soil bulk density | SBD | −0.869 | 0.068 | 2.75 | |
| Soil noncapillary porosity | SNP | 0.908 | 0.254 | 2.89 | |
| Soil capillary porosity | SCP | 0.913 | 0.232 | 2.90 | |
| Soil total porosity | STP | 0.945 | 0.251 | 3.01 | |
| Soil water content | SWC | −0.155 | 0.882 | 1.36 | add in |
| Soil field capacity | SFC | 0.898 | 0.163 | 2.85 | |
| Soil total nitrogen | TN | 0.895 | 0.009 | 2.83 | |
| Soil total phosphorus | TP | 0.954 | 0.074 | 3.02 | add in |
| Soil total potassium | TK | 0.886 | −0.28 | 2.83 | |
| Soil organic carbon | SOC | 0.936 | 0.246 | 2.98 | |
| Soil available nitrogen | AN | 0.854 | −0.183 | 2.71 | |
| Soil available phosphorus | AP | 0.92 | −0.306 | 2.94 | add in |
| Soil available potassium | AK | 0.913 | −0.017 | 2.88 | |
| Soil pH | pH | 0.26 | −0.904 | 1.54 | add in |
| Characteristic root | | 9.983 | 2.079 | | |
| Variance contribution rates/% | | 71.308 | 14.850 | | |
| Accumulated variance contribution rates/% | | 71.308 | 86.159 | | |

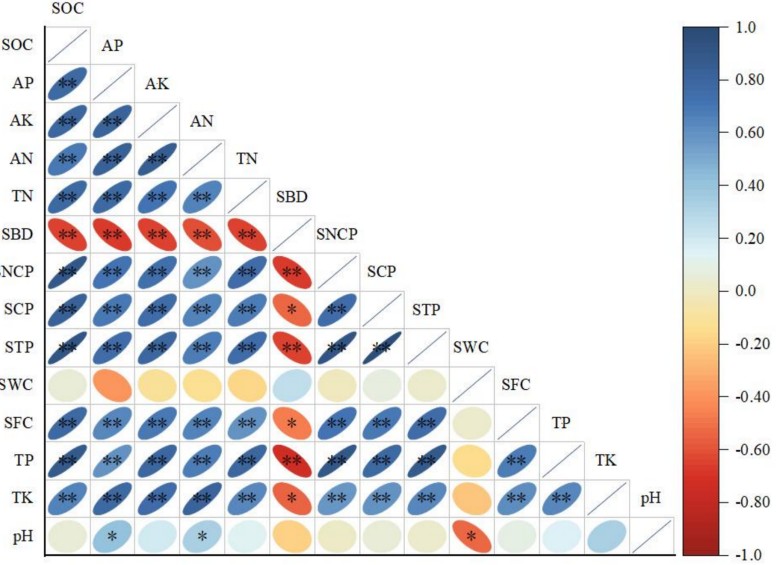

**Figure 5 Analysis of correlation between forest age and soil properties.** Note: One asterisk (*) indicates a significant correlation at the 0.05 level. Two asterisks (**) indicate an extremely significant correlation at the 0.01 level. SBD, Soil bulk density; SNP, Soil noncapillary porosity; SCP, Soil capillary porosity; STP, Soil total porosity; SWC, Soil water content; SFC, Soil field capacity; TN, Soil total nitrogen; TP, Soil total phosphorus; TK, Soil total potassium; SOC, Soil organic carbon; AN, Soil available nitrogen; AP, Soil available phosphorus; AK, Soil available potassium.

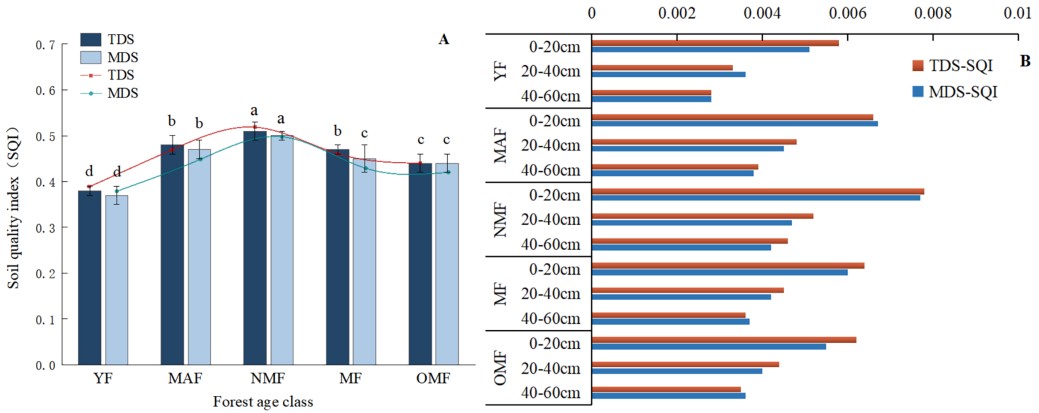

**Figure 6 SQI of the TDS and MDS.**

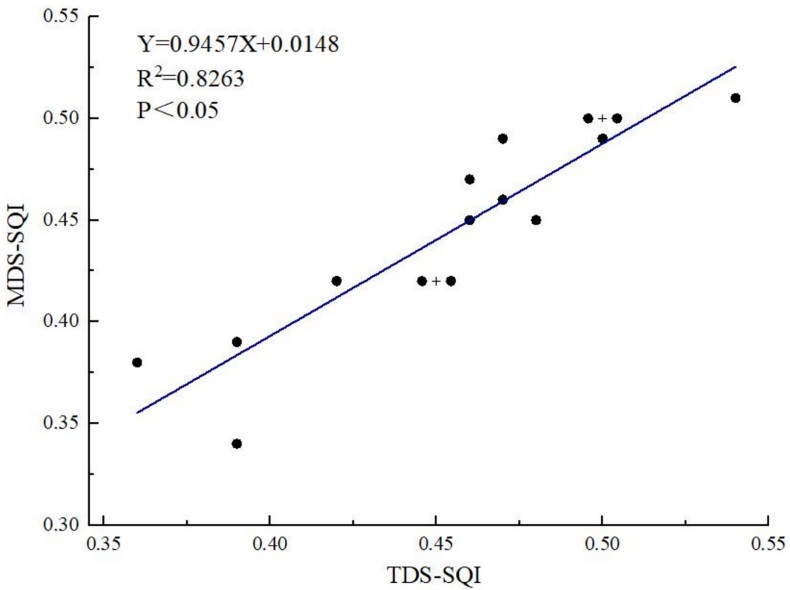

**Figure 7 Linear relationship between the SQI values of the TDS and MDS.**

## Calculation of the SQI

The soil quality index was represented by TDS-SQI for the total data set and MDS-SQI for the minimum data set (Fig. 6A). The values of TDS-SQI ranged from 0.38 to 0.51 and the values of MDS-SQI ranged from 0.37 to 0.50. The pattern of change of TDS-SQI was NMF (0.51) > MAF(0.48) > MF(0.47) > OMF(0.44) > YF(0.38) and the pattern of change of MDS-SQI was NMF(0.50) > MAF (0.47) > MF(0.45) > OMF(0.44) > YF(0.37). The two evaluation results were consistent. By calculating the TDS-SQI and MDS-SQI at different soil depths (Fig. 6B), we see that the results of the soil fertility quality assessment at different soil depths are still NMF > MAF > MF > OMF > YF.

*Reliability tests for minimum data sets*

Regression analyses of the TDS-SQI *versus* the MDS-SQI were performed to verify the reliability of the MDS selection by means of scatter plots of the SQI for three different sites in the same age class. By assessing the fit of the linear regression analysis (Fig. 7) we see that TDS-SQI was significantly correlated with MDS-SQI ($P < 0.05$, $R^2 = 0.8263$). This indicates that the MDS established in this study, is a better alternative to TDS in responding to soil fertility quality in the study area.

# DISCUSSION

## Analysis of changes in soil characteristics indicators at different soil depths

In the 0–60 cm soil depth of the shelter forests studied in this article, soil bulk density and water content increased with soil depth. Soil porosity, field water holding capacity, organic carbon, pH, and nutrient content decreased with soil depth. Many scholars have observed similar patterns in their studies of forest age and soil properties in artificial and shelter forests (*Jiang et al., 2013*; *Wu et al., 2010*; *Yesilonis et al., 2016*). The low water content of the top soil layer of the *Populus simonii* shelter forests in all age classes is mainly due to the high wind speed and strong solar radiation in the Horqin Sandy Land; the top soil layer evaporates faster due to the wind, temperature, and humidity in the surface environment. On the one hand, elements and organic matter in the surface soil mainly come from the decomposition of litter, which increases as the forest ages. The surface humus and plant root growth have a compounding effect. As the porosity of the surface layer of the soil increases the soil bulk density decreases and the soil's permeability, air permeability, and storage capacity increases. In this way the soil water-holding properties improve. On the other hand, the dense distribution of vegetation roots in the soil surface layer and the growth, extension, and aging of the root system have reduced the soil compactness and improved the soil porosity to a certain extent. The deeper soil is subjected to upper loading so the bulk of the soil is more compact in the deeper layer. This means the soil bulk density is greater in the deeper soil, meaning that soil nutrients are usually higher in the surface layer (*Gao et al., 2021*).

## Analysis of changes in soil indicators and fertility quality at different forest ages

After afforestation in sandy soil, the growth of surface herbaceous vegetation, the accumulation of surface litter, and changes in the degree of cover over the years have led to corresponding changes in the microenvironment of the soil. With the increase in the number of years of afforestation, the distribution area of the vegetation root system in the soil has become wider and more active and the physico-chemical properties of the soil have changed accordingly. Litter accumulated on the soil leads to less direct sunlight on the soil and provides more thermal insulation. This reduces the evaporation of surface water and organic matter decomposition. The content of the organic matter and water in the soil, porosity, and bulk density is closely related to the soil organic matter itself, which has a

loose porous structure. This not only promotes the formation of soil aggregates but also improves the soil aeration and water absorption capacity of the soil (*Chen et al., 2021a*). The physical properties of the sandy soil changed and major nutrient elements of the soil increased. This led to a replenishment of the accumulated nutrients in the topsoil in the original sandy soil (*Gao & Huang, 2020*; *Jiang et al., 2013*). Vegetation and soil play a complementary and positive role in the early stages of afforestation. However, when the age of the forest increases the canopy closure of the forest changes and results in an overly large canopy closure. This means that the underlying low vegetation does not receive a reasonable distribution of the ecological resources. Sunlight and rainfall are intercepted, which affects the photosynthesis and growth and development of the understory vegetation. In addition, the newborn seedlings that occur with natural succession do not get enough space to grow. Vegetation and individual trees compete for water and nutrients in the soil (*Guo et al., 2022*). *Populus* itself has a well-developed root system and a strong water-absorbing capacity. For the Horqin Sandy Land, which has an arid climate and infertile soil, the forest is too old which limits the understory biodiversity. Thus the soil quality begins to decline and the forest land begins to degrade.

## Recommendations and outlook

As afforestation time increases, we see that the establishment of sand fixation forests strengthens the soil formation process of sandy soil and improves soil fertility. This makes the moving sand gradually become fixed or semi-fixed sand, alleviating the local dust storms. However, people often neglect the care and management of sand-fixation forests. The results of our study on all age groups of poplar shelterbelts show that sand shelterbelts also need to be nurtured and managed at the right time to be sustainable. It is recommended that the study area carries out the corresponding artificial care measures for *Populus simonii* shelter forests of more than 30 years of age and reasonably applies fertilizers to improve the quality of the forest land soil.

With the world's increasing awareness of the importance of shelterbelts, more studies have been published on the subject. The top three countries for research on shelterbelts are China, the United States, and Canada (*Mayrinck et al., 2019*). The United States has extensive experience in reforesting sandy areas but their shelterbelts have long been subject to human interference (*Ghimire et al., 2014*). The type of shelterbelts in Canada are mainly farmland shelterbelts (*Mayrinck et al., 2019*) and according to the Government of Canada farmers in Western Canada have planted more than 600 million trees over the past century (*Piwowar, Amichev & Van Rees, 2016*). China is the country that has done the most research on shelterbelt forests in sandy areas but researchers have been more concerned with the effect of silvicultural density and the existing studies mainly focus on a few stand ages or a few soil indicators (*Cao, 2008*; *Liu et al., 2023*). We analyzed the soil properties of sandy shelterbelt forests in their natural state for all age groups. The fertility of soils at different growth stages of *Populus simonii* shelter forests when there was no continuous human interference or maintenance is clarified in this study. This is an important reference for the management of poplar shelterbelts in arid and semi-arid areas.

## CONCLUSIONS

*Populus simonii* shelter forests at the southern edge of the Horqin Sandy Land that are growing from young to nearly mature forests (forest age ≤30 a) exhibit soil physico-chemical properties that are developing in a positive direction as the forest ages. This positive direction includes a gradual decrease in soil bulk density and a progressive increase in soil nutrient content, water content, and water holding capacity. During the growth stage from near-mature to over-mature forest (30 a < stand age ≤ 42 a) the soil develops from alkaline to neutral as the stand age increases. The change in soil bulk density is not significant. The soil water content and water holding capacity is reduced to varying degrees and the soil fertility decreases. The evaluation results of soil fertility quality in the total data set of this article were NMF(0.51) > MAF(0.48) > MF(0.47) > OMF(0.44) > YF (0.38), and those in the minimum data set were NMF(0.50) > MAF(0.47) > MF(0.45) > OMF(0.44) > YF(0.37). The two evaluation results were consistent. The results of the two evaluation systems were significant and positively correlated ($P < 0.05$, $R^2 = 0.8263$), which indicated that it was feasible to evaluate the quality of soil fertility of shelter forests of different ages using the minimum data set.

### Funding
This work was supported by the Inner Mongolia Autonomous Region Science and Technology Major Project (No. 2019ZD003-2), the Inner Mongolia Autonomous Region Natural Science Foundation (No. 2021MS03055) and the Inner Mongolia Autonomous Region Science and Technology Programme (No. 2021GG0070). The funders had no role in study design, data collection and analysis, decision to publish, or preparation of the manuscript.

### Grant Disclosures
The following grant information was disclosed by the authors:
Inner Mongolia Autonomous Region Science and Technology Major Project: 2019ZD003-2.
Inner Mongolia Autonomous Region Natural Science Foundation: 2021MS03055.
Inner Mongolia Autonomous Region Science and Technology Programme: 2021GG0070.

### Competing Interests
The authors declare that they have no competing interests.

### Author Contributions
- Xinyu Guo conceived and designed the experiments, performed the experiments, analyzed the data, prepared figures and/or tables, and approved the final draft.
- Guang Yang conceived and designed the experiments, authored or reviewed drafts of the article, and approved the final draft.
- Ji Wu conceived and designed the experiments, performed the experiments, prepared figures and/or tables, and approved the final draft.

- Shi Qiao performed the experiments, analyzed the data, prepared figures and/or tables, and approved the final draft.
- Li Tao performed the experiments, authored or reviewed drafts of the article, and approved the final draft.

## Data Availability

The raw measurements are available in the Supplemental File.

## Supplemental Information

Supplemental information for this article can be found online at http://dx.doi.org/10.7717/peerj.17512#supplemental-information.

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
