# Peer review of "Impacts of forest age on soil characteristics and fertility quality of *Populus simonii* shelter forest at the southern edge of the Horqin Sandy Land, China"

_PeerJ, doi:10.7717/peerj.17512_

## Round 0.1 · original submission · Major Revisions

Please take into all the comments by the reviewers and revise the manuscript or provide a strong justification if you don't agree with certain comments.

**Language Note:** The review process has identified that the English language must be improved. PeerJ can provide language editing services - please contact us at [email protected] for pricing (be sure to provide your manuscript number and title). Alternatively, you should make your own arrangements to improve the language quality and provide details in your response letter. – PeerJ Staff

Reviewer 1 ·

Basic reporting

1. the idea of this paper is clear, and try to find age effects.
2. many new references have been published for shelterbelt forest effects on undergound soils in these years. However, this ms did not mentioned it at all. Please add new references of recent published.
some basic conclusion in NE China plain are, shelterbelt decreased soil water content, indicating high water utlization by poplars. Shelterbelts increased SIC content, thus carbon sequestration. Shelterbelt decreased Q10 value of Rh, thus, increased SOC retention time in the soils and increased SOC sequestration rate. Shelterbelt altered soil nutrients but differed from different N and P fractions. Shelterbelt increased soil porosity by soil aggregating. This basic advances should be disucssed in introduction and get you hypothesis out.
3. It is better to share the raw data without any restriction.
4.I cannot find a hypothesis of this paper. Please try to make a hypothesis-driven paper.

Experimental design

1.According to the regulation of shelterbelt afforestration of China, shelterbelt belts should be strictly protected. If the old forest be cutted, the new shelterbelt should be afforested at the same time and same location. So, I am not sure how to find different aged shelterbelt at the same place?
if you find it, did you try to identify the history of the location? if previous history of this site is poplar, it means legacy effect is there. Or else, if the location is a new afforested site, it means that prevous land use(agriculture or grassland) will greatly affect the results.
At the moment, the author did not give a full description of this in your M&M. This greatly weak your strongthen of this manuscript.
Please give a full description of your criteria of forest age site selection.
Also, in the discussion, please give an uncertainty analysis of this kind of legacy effects.
2. Please provide some pictures of the study sites.
I cannot image how can you find so big shelterbelt forests at 30m*30m. Usually, the shelterbelt forest around farmland is several row line-type forests.
3.2.2.2 should be described in more details on your method. It is difficult to image a standard in Chinese for an international reader, so difficult to judge the method.

Validity of the findings

shelter forests is functioning as its shelter role. I am wondering the shelter role of your finding. It means that this should be discussion by comparing with published paper in China and other world in the same latitude regions, such as Canada.
Please add a table making comparision of your results with other authors, to highlight the importance of your data to fill knowledge gap of previous publication. This is important for publication of your paper.

Reviewer 2 ·

Basic reporting

The English Grammar should be improved greatly.

Experimental design

Materials & Methods
L135: The sampling method is a common method, thus describe it with text. The figure 1 is not necessary.
L143: which indicators were measured and which methods were applied to determine them? Please provide this information in this section rather than showing in Table 2.

Validity of the findings

All underlying data have been provided. In my opinion, this conclusion should be resummarized.

Additional comments

“Impacts of forest age on soil characteristics and fertility quality of Populus simonii shelter forest at the southern edge of the Horqin Sandy land,China”. In this study, the authors evaluated the effects of forest stand age on soil characteristics and fertility quality of Populus simonii shelter forests. They found that soil bulk density and pH decreased with stand age, soil total porosity, organic carbon content increased with stand age. The soil fertility reaches the peak when the forest is nearly mature, and the soil fertility declines after the age of the forest reaches 30 year. These findings provide a scientific basis for soil nutrient regulation and forest management in this region. However, there were some defects, which need to be improved greatly, especially in English grammar.

In introduction
L57: “this study area of this paper” was written by Chinese mode, please rewrite it.

Materials & Methods
L135: The sampling method is a common method, thus describe it with text. The figure 1 is not necessary.
L143: which indicators were measured and which methods were applied to determine them? Please provide this information in this section rather than showing in Table 2.

In section of 2.2 Experimental design, is the formulas of calculating water storage capacity invented by authors or cited from other studies? Please specify them. Moreover, in this formulas, the volume is the volume of litter including saturation water (fresh litter), not that of dry litter. Please provide the related evidence.

Results
In Figure 2, 3 and 5, the more important information is the effect of forest ages, not soil layers. Therefore, the same indicator (soil bulk density, …) in the same layer with different stand ages should be present in the same unit. For example, in figure 2A, the x axis should be soil layers.

Reviewer 3 ·

Basic reporting

This study on how forest age impacts soil fertility is interesting, but the manuscript is not well written, the English language needs to be improved throughout, especially punctuation (e.g. several examples of missing 'space' or writing ',' instead of '.', making it difficult to read. It is sometimes unclear what the authors mean, and there are lots of small mistakes that could have easily been corrected with a simple spelling check before submitting.

References look relevant, but most of them seem to be from Chinese studies; the manuscript would benefit from examples/citations from other parts of the world too. Some sections lack references in the text. The reference list is not in alphabetical order, some references are not cited in the text. There are two 'Chen et al 2021', they should be 'a' and 'b', it is not currently possible to differentiate between them in the text.

Tables and figures are overall well made. Raw data has been shared in supplementary material.

Experimental design

The experimental setup looks sound, the study seems well executed, the raw data looks solid, but the methods need further clarification. How were the soil indicators selected? The 'Minimum Data Set' is not explained enough (especially confusing in the abstract), what was the purpose of it, which indicators were included?

Validity of the findings

The results are interesting and well presented, with statistically significant differences clearly presented.

Additional comments

Detailed comments:
L27: The minimum data set is mentioned in the abstract, but not explained, confusing.
L28 and in many places throughout the manuscript: ‘Space’ missing
L61: Seems to be a comma instead of full stop. Difficult to read.
L67: Water holding capacity
L67: “in this”, unclear what you mean
L89-105: References missing
Section 2.1: A map showing the study area would be good
L 115: “Soil types mainly include non-zonal wind-sand soils and zonal chestnut-calcium soils”. Ref needed, is this a local soil classification? Please also use international classification such as FAO.
L 123: “(LY/T 2908-2017)” this and other mentioned classifications need to be properly cited and included in the reference list.
Section 2.2.2: Rephrase, make sure you use past tense in your methods.
L146: What is understorey soil?
L161: “Thirteen soil indicators…” What were they, how did you define them? Please include a reference here.
L167-170: Not written in past tense
Section 3: Difficult to differentiate between text and sub headings, hard to read.
L216-217: Rephrase, water content is not weak.
L253: “TP”, Explain the abbreviations the first time you use them in the text, not enough just in the figure text.
L 281: “increased with the depth of the soil depth, and soil porosity, field capacity, organic carbon, pH, and nutrient content decreased with the depth of the soil depth.” Remove 'the depth of the'

---

## Round 0.2 · accepted · Accept

Given that the previous reviewers were occupied and could not review the manuscript, I assessed the revisions myself and I am satisfied with the quality of the revised manuscript.